# JANET: Joint Adaptive predictioN-region Estimation for Time-series

## Abstract

Conformal prediction provides machine learning models with prediction sets that offer theoretical guarantees, but the underlying assumption of exchangeability limits its applicability to time series data. Furthermore, existing approaches struggle to handle multi-step ahead prediction tasks, where uncertainty estimates across multiple future time points are crucial. We propose JANET (**J**oint **A**daptive predictio**N**-region **E**stimation for **T**ime-series), a novel framework for constructing conformal prediction regions that are valid for both univariate and multivariate time series. JANET generalises the inductive conformal framework and efficiently produces joint prediction regions with controlled $K$-familywise error rates, enabling flexible adaptation to specific application needs. Our empirical evaluation demonstrates JANET's superior performance in multi-step prediction tasks across diverse time series datasets, highlighting its potential for reliable and interpretable uncertainty quantification in sequential data.

## 1 Introduction

In this work, we tackle the challenging problem of multi-step uncertainty quantification for time series prediction. Our goal is to construct joint prediction regions (JPRs), a generalisation of prediction intervals to sequences of future values. The naive approach of taking the Cartesian product of marginal prediction intervals, each with the desired coverage level $(1 - \epsilon)$, does not guarantee the desired global coverage. If future time steps were independent, this approach would result in coverage $(1 - \epsilon)^H$, where $H$ is the length of the horizon. Although techniques such as Bonferroni correction (13) can adjust for this, it becomes overly conservative as $H$ increases, especially in the presence of temporal dependencies, which, of course, is to be expected in the case of time series data.

Conformal Prediction (CP) (3; 39; 17) offers a distribution-free frequentist methodology for uncertainty quantification with finite-sample coverage guarantees. Specifically, CP ensures that $\Pr(Y \in \mathcal{C}(X^*)) \geq 1 - \epsilon$, where $\mathcal{C}(X^*)$ is the prediction set (or interval in the univariate case) for a new input $X^*$. The theoretical guarantees and model-agnostic nature of CP have spurred its application in diverse areas, including large language models (30) and online model aggregation (19). CP is readily applicable to any algorithm that provides a (non-)conformity score, a measure of how well a data point aligns with the rest of the dataset. The key assumption is exchangeability, which is satisfied for independent and identically distributed (IID) data. Extensions of CP exist for dependent data settings (9; 6; 43), but primarily focus on single-step predictions, and the extension to multistep prediction is not trivial.

CP offers finite-sample guarantees, making it an attractive approach to building joint prediction regions (JPRs). However, the exchangeability assumption of CP is always violated in time series data. Moreover, the multivariate residuals ($H$ in total — one residual per time step) in a multi-step ahead prediction pose a challenge, as for most regression cases, CP relies on the exchangeability of the scalar non-conformity scores and the quantile inversion for calibration. This poses the question of mapping multivariate residuals to a scalar value. Whilst single-step prediction allows for simple residual sorting and the necessary quantile inversion, this approach fails for multi-step scenarios due to the multidimensionality of the residuals. We address the exchangeability issue in the inductive conformal setting by showing an extension of the work of (9) from the transductive (full) to the inductive (split) CP setting. Full CP requires fitting many models, whilst inductive CP only requires fitting a single model. The approach of (9) leverages specific index permutations of a single time series, providing a

distribution of residuals for each permuted time series. We further introduce a novel non-conformity score designed to map multi-dimensional residuals to univariate quantities, enabling the construction of valid JPRs. The resulting framework, JANET (**J**oint **A**daptive predictio**N**-region **E**stimation for **T**ime-series), allows for inductive conformal prediction of JPRs in both multiple and single time series settings. When applied to multiple independent time series, JANET guarantees validity as the permutation across independent time series is exchangeable, whilst, for single time series, it provides approximate validity under weak assumptions on the non-conformity scores as long as transformations (permutations within a single series) of the data are a meaningful approximation for a stationary series.

Our key contributions are: (i) formally generalising the framework in (9) to the inductive setting for computational efficiency whilst maintaining approximate coverage guarantees; (ii) design of non-conformity measures that effectively control the $K$-familywise error (a generalisation of the familywise error) (25; 41), whilst accounting for time horizon and historical context; and (iii) empirical demonstration of JANET's (finite-sample) coverage guarantees, computational efficiency, and adaptability to diverse time series scenarios.

## 2 RELATED WORK

**Relaxing exchangeability**   Recent research has shown a growing interest in adapting conformal prediction (CP) to non-exchangeable data. Early work by (40) explored relaxing the exchangeability assumption using Mondrian CP, which divides observations into exchangeable groups. (14) built upon this idea to share strengths between groups in hierarchical modelling. (37) and (29) addressed covariate and label shifts, respectively, by reweighting data based on likelihood ratios. Similarly, (27) and (8) applied reweighting for predictive inference in causal inference and survival analysis, while (16) focused on controlling covariate shift by statisticians. However, these methods address heterogeneity rather than the serial dependence found in time series.

**One-step or multivariate prediction**   (20) tackled distribution shifts in an online manner by adapting coverage levels based on comparisons of current coverage with desired coverage. (44) extended this work with online expert aggregation. (21) later introduced an expert selection scheme to guide update step sizes. These works typically require a gradient-based approach to learn a model that adapts to the coverage. (6) generalised pure conformal inference for dependent data, using fixed weights for recent training examples to account for distributional drift. (43) employed predictor ensembling, assuming exchangeability but with asymptotic guarantees. (9) leveraged randomisation inference (32) to generalise full conformal inference to serial dependence, achieving valid coverage under exchangeability and approximate validity with serial dependence. Notably, these methods primarily focus on single-step predictions for univariate series. A similar extension to the inductive setting is provided in (12) for the functional time series setting. They randomly selected the time indices to form a calibration series, however, such an approach does not preserve the statistical properties of the time series. Additionally, the work is primarily based on applying to the time series and lacks formalisation as the generalised inductive conformal predictors. In contrast, we split the sequence into two such that we preserve the statistical properties of the time series; our method formally extends (9) to inductive conformal prediction and multi-step scenarios, including multivariate time series.

Other notable works, (11; 1; 42) constructed prediction bands for multivariate functional data, functional surfaces and ellipsoidal regions for multivariate time series, respectively, but only for single-step predictions.

**Multi-step prediction**   (2) applied the jackknife method to RNN-based networks for multi-step prediction, with theoretical coverage of $1 - 2\epsilon$ at significance level $\epsilon$. (33) assumed conditional IID prediction errors in multi-output models, using Bonferroni correction (22) to achieve desired coverage. However, their approach can be overly conservative with increasing prediction horizons and may not hold when model assumptions are violated. Additionally, they require multi-output prediction models. (10) utilised linear complementary programming and an additional dataset to optimise the parameters of a multi-step prediction error model. In recent work, (36), building on (28), employed copulas to adjust for temporal dependencies, enabling multi-step and autoregressive predictions for multivariate time series filling in the deficit from other works. However, their method

requires two calibration sets and gradient-based optimisation, making it data-inefficient. Furthermore, unlike our proposed framework JANET, their approach requires multiple independent time series (just like the preceding works) and cannot adapt prediction regions based on historical context.

A concurrent work (45) primarily focussed on single-step prediction adapted (20) to account for heterogeneous trajectories in multiple time series settings. However, they require multiple calibration sets and do not show how to control $K-$familywise error when dealing with multi-step prediction.

**Non-conformal prediction regions**  Beyond conformal methods, bootstrapping provides an alternative method for constructing joint prediction regions especially when one only has a single time series. (34) generates $B$ bootstraps and finds a heuristic-based prediction region from the bootstrapped predictive sequences, however, the method only provides have asymptotic guarantees. (41) creates JPRs that are asymptotically valid and can control the $K$-familywise error. However, given that this method is based on bootstrapping, it has a large computational cost that is infeasible when working with neural networks. Our work can be seen as conformalisation of (41; 15) without relying on bootstrapping and with adaptive prediction regions. Additionally, as a conformal method, JANET can be readily applied to time series classification tasks which is not apparent for bootstrap-based methods.

To the best of our knowledge, JANET is the only framework that can handle both single and multiple time series (univariate or multivariate), whilst providing adaptive prediction regions based on historical context (lagged values), and controlling $K$-familywise error with finite-sample guarantees.

## 3  BACKGROUND ON CONFORMAL PREDICTION

In this section, we provide an overview of conformal prediction for IID data in the context of regression tasks. In the next section, we specify CP for the time series setting.

Full conformal predictors assess the *non-conformity* of a test sample $x^*$ and a postulated target $y$ to a training set by running the underlying algorithm $(n + 1)c$ times, where $n$ is the number of training samples and $c$ is the number of points along a discretised grid of the target space ($\mathcal{Y}$). This is done to construct prediction sets by inversion: the set of grid points that best conform according to a desired significance level $\epsilon$. Full CP is prohibitive for compute-intensive underlying training algorithms as it requires refitting a model for each postulated label/grid point. Inductive conformal predictors (ICPs) offer an elegant solution to this problem by training the model only once. We focus on regression tasks with ICPs.

Let $\mathbb{P}$ be the data generating process for the sequence of $n$ training examples $Z_1, \ldots, Z_n$, where for each $Z_i = (X_i, Y_i)$ pair $X \in \mathcal{X}$ is the sample and $Y \in \mathcal{Y}$ is the target value. We partition the sequence into a proper training set, $Z_{tr} = \{Z_1, \ldots, Z_{n_{tr}}\}$ (the first $n_{tr}$ elements), and the remaining $n_{cal}$ elements form a calibration set, $Z_{cal} = \{Z_{n_{tr}+1}, \ldots, Z_{n_{tr}+n_{cal}}\}$, such that $n = n_{tr} + n_{cal}$. A point prediction model, $\hat{f}$, is trained on the training set proper $Z_{tr}$, and non-conformity scores, $a_i$, are computed for each element of the calibration set, $Z_{cal}$.

In standard regression, a natural non-conformity score is the absolute residual $a_i = |y_i - \hat{f}(x_i)|$. By construction, for any $i$, $y_i \in [\hat{f}(x_i) \pm a_i]$; $a_i$ is half the interval width that ensures coverage for any new sample (assuming symmetric intervals). In ICP we compute a non-conformity score for each of the $n_{cal}$ samples in the calibration set and then sort them from largest to smallest. Let $a_{(1)}$ denote the largest and $a_{(n_{cal})}$ denote the smallest. Then intervals of the form $(\hat{f}(x_i) \pm a_{(1)})$ will cover all but one sample from the calibration set and intervals of the form $(\hat{f}(x_i) \pm a_{(n_{cal})})$ will cover none of the samples from the calibration set (assuming no ties).

Extending this line of thinking, a prediction interval with $(1 - \epsilon)$ coverage can be obtained by inverting the quantile of the distribution of non-conformity scores $a_i$. To do so we find the $\lfloor \epsilon(n_{cal} + 1) \rfloor^{th}$ largest non-conformity score, $q_{1-\epsilon}^a := a_{(\lfloor \epsilon(n_{cal}+1) \rfloor)}$ and set this to half of our interval width. For an unseen $Z^* = (X^*, Y^*)$, we will provide a prediction interval of the form

$$\mathcal{C}_{1-\epsilon}^a(X^*) = \left( \hat{f}(X^*) - q_{1-\epsilon}^a, \hat{f}(X^*) + q_{1-\epsilon}^a \right). \tag{1}$$

A drawback of this method is that the intervals are of constant width, $2q_{1-\epsilon}^a$. (26) suggest an alternative non-conformity score that takes into account local information, $r_i = \frac{|y_i - \hat{f}(x_i)|}{\hat{s}(x_i)}$. Here, $\hat{s}(\cdot)$ is also fit on the train set and predicts the conditional mean absolute deviation. Now we can provide locally adaptive prediction intervals for a test sample $Z^* = (X^*, Y^*)$ with $(1 - \epsilon)$ coverage:

$$\mathcal{C}_{1-\epsilon}^r(X^*) = \left( \hat{f}(X^*) - q_{1-\epsilon}^r \cdot \hat{s}(X^*), \hat{f}(X^*) + q_{1-\epsilon}^r \cdot \hat{s}(X^*) \right) \tag{2}$$

where $q_{1-\epsilon}^r$ is the $\lfloor \epsilon(n_{cal} + 1) \rfloor^{th}$ largest non-conformity score. In Eq. (1), the $a_i$ (and $q_{1-\epsilon}^a$) directly define the width of the interval whilst in Eq. (2), the $r_i$ (and $q_{1-\epsilon}^r$) inform how much to rescale the conditional mean absolute deviation to achieve coverage of a particular level.

## 4 GENERALISED INDUCTIVE CONFORMAL PREDICTORS

In this section, we formally describe a generalised framework for inductive conformal predictors based on Generalised CP (9) and randomisation inference (32) applied to time series forecasting. With minor notational adjustments, our generalised ICP extends to multi-step and multivariate time series prediction. In the following sections, we demonstrate how to obtain exact validity guarantees for independent time series (or IID data) and approximate validity guarantees for a single time series. Our task is to forecast $H$ steps into the future conditioned on $T$ steps of history.

### 4.1 MULTIPLE TIME SERIES

Assume we have $n$ independent time series $\mathbf{Z} = \{Z_k\}_{k=1}^n$ where each individual time series, $Z_k$, is an independent realisation from an underlying distribution $\mathbb{P}$ (and within each $Z_k$ there is temporal dependence). As usual in ICP, we split $\mathbf{Z}$ into a proper training sequence $\mathbf{Z}_{tr} = \{Z_k\}_{k=1}^{n_{tr}}$ and a calibration sequence $\mathbf{Z}_{cal} = \{Z_{n_{tr}+i}\}_{i=1}^{n_{cal}}$. Without loss of generality, we assume the time series are length $T + H$. For each time series, we can define $Z_k = \{z_{k,1}, z_{k,2}, \ldots, z_{k,T+H}\}$ as $(X_k, Y_k)$, each with $X_k = \{z_{k,1}, \ldots z_{k,T}\} = \{x_{k,1}, \ldots x_{k,T}\}$ denoting the relevant series' history, $Y_k = \{z_{k,T+1}, \ldots, z_{k,T+H}\} = \{y_{k,1}, \ldots, y_{k,H}\}$ being the values at the next $H$ time steps and each $z_{k,j} \in \mathbb{R}^p$ (where $p = 1$ corresponds to univariate time series and $p > 1$ corresponds to multivariate time series). In other words, each time series can be split into the history we use to predict ($X_k$) and the target ($Y_k$). As in any other ICP setting, we can train a model $\hat{f} : \mathbb{R}^{p \times T} \to \mathbb{R}^{p \times H}$ that predicts $H$ steps into the future based on $T$ steps of history. Then, we can compute non-conformity scores for a non-conformity scoring function $\mathcal{A}$. Note that we can form a distribution over non-conformity scores with one non-conformity score per time series. From there we can invert the quantiles and produce a prediction interval.

### 4.2 SINGLE TIME SERIES

Unlike the case of multiple time series, we only have a single time series and we do not have access to an entire distribution of non-conformity scores—we only have a single score. To address this problem we apply a permutation scheme from (9) on the calibration series that provides a distribution over the conformity scores, this is equivalent to the view of randomisation inference employed in (9). We compute the $p$-values as in Definition 1. Now we can treat this collection of permuted time series in the same way we did for multiple time series. Each permutation will provide a non-conformity score and this is how we approximate a distribution of non-conformity scores.

We now assume we have a single time series $Z = \{z_1, \ldots z_L\}$ where $L = L_{tr} + L_{cal}$ is the length of the entire *single* series, $L_{tr}, L_{cal} \geq T + H$ and all $z_i \in \mathbb{R}^p$. We split $Z$ into a train subseries, $Z_{tr} = \{z_1, \ldots, z_{L_{tr}}\}$, and a calibration subseries, $Z_{cal} = \{z_{L_{tr}+1}, \ldots, z_{L_{tr}+L_{cal}}\}$. $Z_{tr}$ is used to fit our point prediction model $\hat{f}$ and $Z_{cal}$ will be used to calibrate our prediction intervals.

Let $\Pi$ be a set of permutations of the indices $\{1, \ldots, L_{cal}\}$. For a permutation $\pi \in \Pi$, let $Z_{cal}^\pi = \{z_{L_{tr}\pi(i)}\}_{i=1}^{L_{cal}}$ denote a permuted version of the calibration sequence, $Z_{cal}$. We call $\mathbf{Z}_{cal}^\Pi = \{Z_{cal}^\pi\}_{\pi \in \Pi}$ the set of permuted version of $Z_{cal}$ under the permutations in $\Pi$. We can partition each permutation of $Z_{cal}$ into $X$ and $Y$ components as in the multiple time series setup where $X$ is the lagged version of $Y$.

**Permutation scheme** (9) introduced a permutation scheme to address data dependence. We adapt this scheme and their notation to our setting on $Z_{cal}$. Given a time calibration sequence of length $L_{cal}$, divisible (for simplicity) by a block size $b$, we divide the calibration series into $d = L_{cal}/b$ non-overlapping blocks of $b$ observations each. The $j^{th}$ non-overlapping block (NOB) permutation is defined by the permutation $\Pi_{j,NOB} : \{1, \ldots, L_{cal}\} \rightarrow \{1, \ldots, L_{cal}\}$ where

$$i \rightarrow \pi_{j,NOB}(i) = \begin{cases} i + (j-1)b & \text{if } 1 \le i \le L_{cal} - (j-1)b \\ i + (j-1)b - l & \text{if } L_{cal} - (j-1)b + 1 \le i \le L_{cal} \end{cases} \quad \text{for } i = 1, \ldots, L_{cal}.$$

Figure 3 in the Appendix provides a visualisation of the NOB permutation scheme to a hypothetical time series $Z$ with $L = L_{tr} + L_{cal} = 10 + 6 = 16$ and $b = 1$.

### 4.3 VALIDITY OF ICP WITH PERMUTATIONS

In Theorem 1 we establish that under mild assumptions on ergodicity and small prediction errors, as defined in A.2.1, we can achieve approximate validity in the case of a single time series whilst using the permutation scheme from (9). In the case of IID data, we have exact validity guarantees shown in Theorem 2 and this exact validity can be applicable in the case of multiple independent time series.

**Definition 1** (Randomised $p$-value). *We define the randomised $p$-value as:*

$$\hat{p} = \hat{p}(y) := \frac{1}{d} \sum_{j=1}^{d} \mathbf{1} \left( \mathcal{A} \left( Z_{tr}, Z_{cal}^{\pi_j} \right) \ge \mathcal{A} \left( Z_{tr}, Z_{cal} \right) \right)$$

*where $\Pi$ is a group of permutations of size $d$ and $\mathcal{A}$ is a non-conformity measure.*

Effectively we are computing an empirical quantile of our test sample's non-conformity relative to the calibration set. Let

$$\alpha_j := \mathcal{A}(Z_{tr}, Z_{cal}^{\pi_j}) \text{ for } j = 1, \ldots, d$$

where $\pi_j$ is the $j^{th}$ permutation of $\Pi$. We can invert the quantile to gain a prediction interval as in Eq. (1). Theorem 1 is adapted from (9) to the inductive conformal setting.

**Theorem 1** (Approximate General Validity of Inductive Conformal Inference). *Under mild assumptions on ergodicity and small prediction errors (see Appendix A.2.1), for any $\epsilon \in (0, 1)$, the approximate conformal p-value is approximately distributed as follows:*

$$|\Pr(\hat{p} \le \epsilon) - \epsilon| \le 6\delta_{1d} + 4\delta_{2m} + 2D \left( \delta_{2m} + 2\sqrt{\delta_{2m}} \right) + \gamma_{1d} + \gamma_{2m} \tag{3}$$

*for any $\epsilon \in (0, 1)$ and the corresponding conformal set has an approximate coverage $1 - \epsilon$, i.e,*

$$|\Pr(y^* \in \mathcal{C}_{1-\epsilon}) - (1 - \epsilon))| \le 6\delta_{1d} + 4\delta_{2m} + 2D \left( \delta_{2m} + 2\sqrt{\delta_{2m}} \right) + \gamma_{1d} + \gamma_{2m}. \tag{4}$$

The proof can be found in Appendix A.2.

**Remark:** (9) demonstrated that conformity scores obtained via the permutations (Figure 3) offer a valid approximation to the true conformity score distribution for strongly mixing time series. Notably, stationary processes like Harris-recurrent Markov chains and autoregressive moving average (ARMA) models exhibit strong mixing properties (5; 4). Statistical tests designed for ARMA models, such as the Ljung-Box or KPSS tests, can be employed to assess the presence of strongly mixing. Our empirical findings suggest that even when stationarity is violated, approximate coverage can still be achieved (see Table 1).

## 5 JANET: JOINT ADAPTIVE PREDICTION-REGION ESTIMATION FOR TIME-SERIES

A joint prediction region (JPR) typically controls the probability of the prediction region containing the entire true prediction sequence at a specified significance level $\epsilon$. This can be interpreted as the

probability of observing at least one element of the true sequence outside the region, also known as the familywise error rate (FWER), namely:

$$\text{FWER} := \Pr(\text{at least one of the } H \text{ components are not in the computed region}).$$

However, as the prediction sequence length, $H$, increases, controlling for FWER can become overly strict and lead to excessively large prediction regions. In such cases, the $K$-FWER offers a valuable generalisation (25; 41). $K$-FWER relaxes the FWER definition, allowing for a specified number of errors ($K$) within the true prediction sequence:

$$K\text{-FWER} := \Pr(\text{at least } K \text{ of the } H \text{ components are not in the computed region})$$

When $K = 1$, the $K$-FWER reduces to the standard FWER. By allowing a tolerance for errors, larger values of $K$ yield smaller, more informative prediction regions, which can be beneficial in decision-making scenarios where some degree of error is acceptable.

**Remark:** JPRs can be constructed in various forms. Whilst hyperspherical construction is a common choice, it may not facilitate reasoning about individual time steps in the horizon. Although it is possible to project the hypersphere onto a hyperrectangle to enable component-wise analysis, this results in a larger region and a loss of predictive efficiency (11; 41).

We introduce JANET (**J**oint **A**daptive predictio**N**-region **E**stimation for **T**ime-series) to control $K$-FWER in multi-step time series prediction.

We adopt the notation for the multiple time series setting and further assume that the time series are univariate (i.e. $p = 1$). In the case of a single time series, we use the same non-conformity scores but treat the permutations of the single time series as distinct, exchangeable time series. We propose two non-conformity measures that are extensions of a locally adaptive non-conformity score from (26)

$$\alpha_i^K := K\text{-}\max\left\{\frac{|y_{i,1} - \hat{y}_{i,1}|}{\hat{\sigma}_1}, \ldots, \frac{|y_{i,H} - \hat{y}_{i,H}|}{\hat{\sigma}_H}\right\} \text{ and} \tag{5}$$

$$R_i^K := K\text{-}\max\left\{\frac{|y_{i,1} - \hat{y}_{i,1}|}{\hat{\sigma}_1(X_i)}, \ldots, \frac{|y_{i,H} - \hat{y}_{i,H}|}{\hat{\sigma}_H(X_i)}\right\} \tag{6}$$

for $i \in \{1, \ldots, n_{cal}\}$ where, $K\text{-}\max(\vec{x})$ is the $K^{th}$ largest element of a sequence $\vec{x}$, $y_{i,j}$ is the $j^{th}$ entry of the target $Y_i$ and $\hat{y}_{i,j}$ is the $j^{th}$ entry of $\hat{f}(X_i)$. Note that the prediction steps can be generated by any model $\hat{f}$ (AR models with $H = 1$ or multi-output models with $H > 1$). The difference between Eqs. (5) and (6) is only in the scaling factors (denominators). In Eq. (5) the scaling factors $\hat{\sigma}_1, \ldots, \hat{\sigma}_H$ are standard deviations of the error, that are computed on the proper training sequence. These scaling factors account for differing levels of variability and magnitude of errors across the prediction horizon but do not depend on the history. Meanwhile, for Eq. (6) the scaling factors are conditional on the relevant history $X$.

We call these functions $\hat{\sigma}_1(\cdot), \ldots, \hat{\sigma}_H(\cdot)$ and they are also fit on the training sequence. Even with IID errors, the predictor may have higher errors for certain history patterns. The history-adaptive conformity score penalises these residuals and aims to deliver uniform miscoverage over the prediction horizon.

**Note:** In the multivariate case (i.e., $z \in \mathbb{R}^p$, $p > 1$), the entries of Eqs. (5) and (6), whose entries are $\left[\frac{|y_{i,j} - \hat{y}_{i,j}|}{\hat{\sigma}_j(X_i)}\right]_j$ for $j \in \{1, \ldots, p\}$. We then take the $K\text{-}\max$, across all $p$ dimensions, and all $H$ steps in the time horizon.

The quantile $q_{1-\epsilon}^\alpha$ can be found by inverting the conformity scores. Then a JPR of the desired significance $\epsilon$ and tolerance $K$ for a test sample $Z^* = (X^*, Y^*)$ (where $Y^*$ is unknown) can be constructed as follows:

$$\mathcal{C}_{1-\epsilon}^\alpha(X^*) = \left(\hat{y}_1 \pm q_{1-\epsilon}^\alpha \cdot \hat{\sigma}_1\right) \times \cdots \times \left(\hat{y}_H \pm q_{1-\epsilon}^\alpha \cdot \hat{\sigma}_H\right) \tag{7}$$

whereas, the JPR for the second score that incorporates historical context are:

$$\mathcal{C}_{1-\epsilon}^R(X^*) = \left(\hat{y}_1 \pm q_{1-\epsilon}^R \cdot \hat{\sigma}_1(X^*)\right) \times \cdots \times \left(\hat{y}_H \pm q_{1-\epsilon}^R \cdot \hat{\sigma}_H(X^*)\right) \tag{8}$$

where $X^*$ is the sequence of lagged values (i.e. the history) for the unknown $Y^*$ that is to be predicted. The $K\text{-}\max$ operation maps the multidimensional residuals to a singular score which allows for

quantile inversion as done in Eq. (1). Further, by taking the $K$-$\max$, each $\alpha_i^K$ (or $R_i^K$) is the value to rescale each $\hat{\sigma}_j$ by to provide intervals that cover all but $K$ of the predicted time points for a specific trajectory $Z_i \in \mathbf{Z}_{cal}$.

We provide two methods for producing JPRs, JANET* and JANET, and describe how to construct a JANET JPR in Algorithm 1:

- **JANET***: Adapts prediction intervals over time horizon as defined in Eq. (7) and only requires fitting a single model.

- **JANET**: Adapts prediction intervals conditional on the relevant history as defined in Eq. (8).

---

**Algorithm 1:** JANET algorithm

---

**Input:** Time series $Z$, significance level $\epsilon$, group of permutations $\Pi = \{\pi_i\}_{i=1}^d$, length of relevant history $T$, prediction horizon $H$, error tolerance $K$

**Output:** Joint Prediction Region (JPR)

**begin**

    1. Partition $Z$ into training sequence $Z_{tr}$ and calibration sequence $Z_{cal}$

    2. Train a prediction model, $\hat{f}(\cdot)$, on the training sequence $Z_{tr}$.

    3. Train an error predicting model $\hat{\sigma}(\cdot)$ on the training sequence $Z_{tr}$.

    4. **for** $i \in \{1, \ldots, d\}$ **do**

        a. Generate permuted calibration series $Z_{cal}^{\pi_i}$

        b. Compute nonconformity score $\alpha_i^K$ according to Eq. (6)

    5. Invert the $\epsilon$-quantile of the set of nonconformity scores $\{\alpha_i\}_{i=1}^d$.

    6. Construct the JPR as defined in Eq. (8).

    **return** JPR

---

**Remark:** The constructed regions are two-sided and symmetric, we discuss the construction of one-sided intervals and asymmetric intervals in Appendix A.1.

# 6 EXPERIMENTS

We demonstrate the utility of our method, JANET, on single time series and multiple time series. For the single time series, we compare against Bonferroni and Scheffé JPRs. Further, we compare against bootstrapping methods. Note that for Bonferroni and Scheffé we can only train the models for complete coverage (1-FWE), whilst for Bootstrap-JPR we can vary $K$ (as with our method). Despite matching this level of flexibility in the choice of $K$, Bootstrap-JPR is much more computationally intensive than JANET. For the multiple time series, we compare against baselines (CopulaCPTS (36), CF-RNN (33), MC-Dropout (18)) that make stronger assumptions (different series being independent) than us on synthetic datasets as well as real-world data. Our generalised ICP method can lead to greater data efficiency whilst creating approximately valid prediction sets should independence be violated.

## 6.1 SINGLE TIME SERIES EXPERIMENTS

We now focus on the scenario where only a single time series is available, and the goal is to construct a JPR for horizon $H$. Common approaches, such as bootstrapping, are used to estimate prediction errors and subsequently compute JPRs. However, these methods suffer from two limitations: (i) bootstrap guarantees often hold only asymptotically, not for finite samples, and (ii) bootstrapping can be computationally expensive, particularly when using neural networks as function approximations (or predictors). We compare JANET to the following baselines:

- **Bonferroni Correction** (13; 22): This approach controls the FWER, but it is conservative. We use bootstrapping-based methods to find the standard deviation of the prediction errors before applying the correction.

- **Scheffé-JPR** (35): This statistical method assumes normality of errors but may not hold for prediction intervals.

- **Bootstrap-JPR** (41; 15): This method is based on bootstrapping and lacks finite-sample guarantees and can be computationally demanding.

We use an ARIMA model as the learner for all methods in this section. Due to computational constraints over numerous simulations, we do not use neural networks for the main coverage experiments. Both Bootstrap-JPR and our proposed method, JANET, control the $K$-FWER (25), so we compare for different tolerance levels, $K$. We want to point out that Bootstrap-JPR can be conformalised in a naive way. Whilst conformalised bootstrapping can address finite-sample issues, it does not reduce computational cost. In contrast, JANET requires training the model only once per simulation. We compare both of our variants: JANET and JANET*.

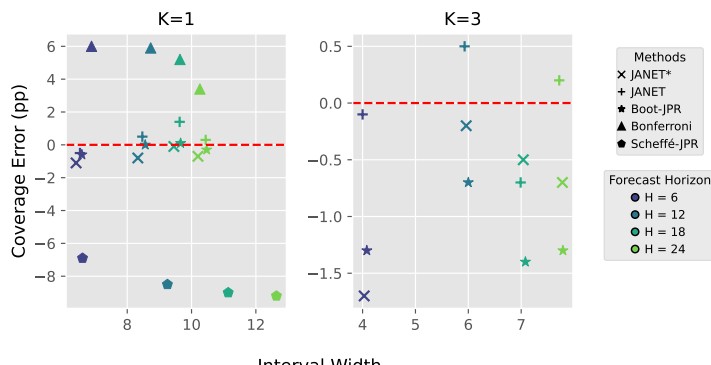

Figure 1: Monte Carlo Experiment Coverage Error (pp) vs Interval Width. The $y$-axis shows the difference in coverage from the target $1 - \epsilon$ for $\epsilon = 0.2$ in percentage points (pp) and the $x$-axis is the geometric mean of the interval width over the time horizon. The red line represents perfect calibration. Better methods can be found near the red line and to the left (well-calibrated, narrower interval width). The left plot is for the case of $K = 1$ whilst the right plot shows $K = 3$. The Bonferroni and Scheffé-JPR methods are only applicable for $K = 1$. Shapes represent calibration methods and colours signify forecast horizon, $H$.

### 6.1.1 MONTE CARLO SIMULATIONS

We generate data from an $AR(2)$ process with $\rho \in \{1.25, -0.75\}$ and evaluate empirical coverage across 1000 simulations. We compute the JPRs for $K \in \{1, 2, 3\}$, $H \in \{6, 12, 18, 24\}$ and significance level, $\epsilon \in \{0.7, 0.8, 0.9\}$. For methods that cannot control the $K$-FWER, the corresponding results refer to $K = 1$. The interval width of one JPR is calculated as the geometric mean of the widths over the horizon $H$. The average over all simulations per setting is reported in Tables 6 to 8.

Tables 3 to 5 in the Appendix present coverage results for $\epsilon$ and varying tolerances $K$, whilst Figures 4 to 6 in the Appendix display coverage errors as bar plots. As expected, Bonferroni is conservative, particularly at larger $\epsilon$. Scheffé-JPR undercovers substantially, likely due to the normality assumption. Bootstrap-JPR and JANET* perform comparably. JANET often has smaller coverage errors but slightly larger average widths, as it treats errors uniformly across the history covariates. Additionally, Tables 6 to 8 in the Appendix present empirical width results for various significance levels $\epsilon$ and varying tolerances $K$ and Figures 7 to 9 in the Appendix plot empirical widths against forecast horizon ($H$) for various significance levels $\epsilon$ and $K = 1$. Figure 1 compares coverage errors against widths, with the ideal method being close to zero error with minimal width within each colour group (representing the same tolerance $K$). Note that within a level of $K$, the points tend to cluster together. (refer to Figure 2 in the Appendix for all different $K$ and all significance levels $\alpha$). The left plot ($K = 1$) includes more points per cluster as Bonferroni and Scheffé-JPR control for this tolerance level only. The analysis also shows that JANET* generally achieves good coverage with minimal width.

### 6.1.2 U.S. REAL GROSS DOMESTIC PRODUCT (GDP) DATASET

We evaluate JANET on the U.S. real gross domestic product (GDP) dataset (38). To address non-stationarity, we log-transform and de-trend the data as a preprocessing step. The resulting series is shown in Figure 10 in the Appendix. Our task is to forecast the next $H = 4$ quarters (equivalent to one year) and construct a JPR for the true sequence. We set the significance level to $\epsilon = 0.2$. Due to the limited availability of real-world data, we create windowed datasets to increase the number of datasets for evaluating coverage. Each window consists of a sequence of 48 quarters (12 years) for JPR computation, followed by a true sequence of 4 quarters (1 year) for coverage assessment. It should be noted that this method for computing empirical coverage has two deficiencies, (i) there are only 100 series (created through windowing); (ii) the series are not independent of each other. Nevertheless, it still provides an assessment of the out-of-sample performance of the method. Table 1 shows the coverage results of the data. Similar to the experiment in Monte Carlo simulations in the previous section, the out-of-sample coverages for JANET and Bootstrap-JPR are close to the desired level at all tolerances $K$, whereas Bonferroni overcovers and Scheffé-JPR undercovers.

Table 1: Empirical Out-of-Sample Coverages on the US GDP Data ($\epsilon = 0.2$, target coverage 80%) and training times (minutes), numbers in the parenthesis refer to values of $K$. Bootstrap-JPR (Boot) and JANET perform similarly whereas Bonferroni (Bonf.) shows over coverage as usual and Scheffé undercovers. Bonf. and Scheffé can only be performed for $K = 1$. The bootstrap methods take approximately 13 times as long as our method in wall-clock time in our implementations.

|            | Bonf.(1) | Scheffé(1) | Boot(1) | **JANET(1)** | Boot(2) | **JANET(2)** | Boot(3) | **JANET(3)** |
|------------|----------|------------|---------|--------------|---------|--------------|---------|--------------|
| Cov. (%)   | 83       | 63         | 78      | 79           | 80      | 78           | 79      | 81           |
| Time (min) | 5        | 8          | 91      | 7            | 91      | 7            | 91      | 7            |

### 6.2 MULTIPLE INDEPENDENT TIME SERIES EXPERIMENTS

In this section, we focus on the setting with multiple independent time series as in (33; 36). As previously noted, this scenario allows us to achieve exact validity guarantees, rather than approximate validity, due to the independence assumption. We compare JANET against the following methods:

- **CF-RNN** (33) is designed for multi-output neural networks–the entire predictive sequence is outputted at once. It assumes conditional IID prediction errors, which may not always be accurate, especially if the trained model has not captured the underlying trend. Further, it relies on the Bonferroni correction (22), which tends to be conservative.

- **MC-Dropout** (18) is a Bayesian method for neural networks that can provide prediction intervals that are often overly narrow intervals with poor coverage, indicating overconfidence.

- **CopulaCPTS** (36) uses copulas to adjust for dependencies. However, it is data-inefficient since it requires two calibration sets and unlike JANET, it cannot adapt its regions based on history or handle different values of $K$ in the $K$-FWER control problem.

- **CAFHT** (45) uses ACI (8) for primarily heterogenous trajectories.. Just like CopulaCPTS, it is also data-inefficient due to additional tuning parameters and cannot adapt its regions based on history or handle different values of $K$ in the $K$-FWER control problem.

Following the evaluation approach used in the baseline methods, we report the "frequency of coverage" on the test set, which should not be interpreted as frequentist coverage, as the latter requires asymptotic repetition (39).

### 6.2.1 PARTICLE SIMULATION DATASETS

We evaluate our model on two synthetic datasets from (24) using the same experimental settings as in (36). In both cases, we predict $H = 24$ steps into the future based on a history of length $T = 35$. Each time step is in $\mathbb{R}^2$. In the two setups, we add in mean-zero Gaussian noise with $\sigma = 0.05$ and $\sigma = 0.01$, respectively. We used two different predictors for the experiments. See Appendix B.2 for predictor model and training details. Table 2 shows results for the *particle5* experiment $\sigma = 0.05$. Under the EncDec predictor, our coverage is closer to the desired level.

For RNN architectures, we observe slight under-coverage for JANET and over-coverage for CF-RNN, whilst other methods, especially MC-dropout, exhibit severe under-coverage. In the *particle1* experiment (Table 2, $\sigma = 0.01$), JANET achieves better coverage under both predictors. Other methods again show significant under-coverage, particularly MC dropout.

### 6.2.2 UK COVID-19 DATASET

We evaluate JANET on the UK COVID-19 dataset with daily case counts from 380 UK regions (33; 36). Table 2 presents the results. The task is to predict daily cases for the next 10 days based on the previous 100 days. Whilst the COVID case sequences from different regions are not independent, we anticipate at least approximate validity using our generalised ICP framework. JANET's coverage is close to the desired significance level $\epsilon$. CF-RNN shows overcoverage with the basic RNN architecture but performs closer to the desired level with the EncDec architecture.

Table 2: Comparison of coverage (%) and interval widths/areas on the test set for $\epsilon = 0.1$. Coverage values closest to 90% are highlighted in bold for every grouping of architecture (RNN, EncDec) on each dataset. Narrower intervals are preferred. Coverages close to the desired significance level for $K = 1$ are bolded.

| | Method | Particle1 | | Particle5 | | UK COVID-19 | |
|---|---|---|---|---|---|---|---|
| | | Coverage | Area | Coverage | Area | Coverage | Width |
| $K = 1$ | MC-Dropout | 79.40 | 2.2026 | 43.40 | 2.1846 | 0.00 | 1969 |
| | CF-RNN | 95.20 | 1.1210 | 95.60 | 6.3749 | 92.50 | 19356 |
| | CopulaCPTS-RNN | **89.60** | 0.9036 | 90.40 | 5.1736 | 85.00 | 16109 |
| | CAFHT-RNN | 93.30 | 1.6913 | 86.60 | 4.6579 | 92.50 | 19356 |
| | JANET-RNN | 85.80 | 0.7372 | **89.80** | 4.7120 | **88.75** | 19054 |
| | CF-EncDec | 98.80 | 8.5731 | **92.40** | 5.8444 | **87.50** | 19194 |
| | CopulaCPTS-EncDec | 85.60 | 2.9410 | 86.40 | 5.3351 | 78.75 | 14572 |
| | CAFHT-EncDec | 94.60 | 1.0053 | 93.00 | 4.9368 | 92.50 | 19356 |
| | JANET-EncDec | **90.60** | 1.0053 | **87.60** | 4.2098 | **87.50** | 19188 |
| $K = 2$ | JANET-RNN | 87.20 | 0.7011 | 89.20 | 4.1417 | 91.25 | 16007 |
| | JANET-EncDec | 90.20 | 0.9018 | 88.00 | 4.3892 | 87.50 | 16319 |
| $K = 3$ | JANET-RNN | 87.00 | 0.6542 | 89.00 | 3.9888 | 91.25 | 14551 |
| | JANET-EncDec | 90.60 | 0.8552 | 87.80 | 4.2098 | 90.00 | 14572 |

## 7 DISCUSSION AND FUTURE WORK

In this paper, we have formally extended the Generalised Conformal Prediction framework proposed by (9) to the inductive conformalisation setting (40). Building upon this foundation, we have introduced JANET (**J**oint **A**daptive predictio**N**-region **E**stimation for **T**ime-series), a comprehensive framework for constructing prediction regions in time series data. JANET is capable of producing prediction regions with marginal intervals that adapt to both the time horizon and the relevant historical context of the data. Notably, JANET effectively controls the $K$-FWER, a valuable feature particularly when dealing with long prediction horizons.

JANET includes several desirable properties: computational efficiency, as it requires only a single model training process; applicability to scenarios involving multiple independent time series, with exact validity guarantees; and approximate validity guarantees when a single time series is available. Furthermore, JANET is flexible enough to provide asymmetric or one-sided prediction regions, a capability not readily available in many existing methods.

Looking towards future work, we envision several promising directions for extending JANET. One avenue involves exploring the extension of JANET to a cross-conformal setting (40), which could offer gains in predictive efficiency at the expense of potentially weaker coverage guarantees. Additionally, we acknowledge a current limitation of JANET, which is its inability to create multiple disjoint prediction regions (23). Such disjoint regions could be denser than a single joint region, thereby providing more informative uncertainty estimates, particularly in cases with multimodal predictive distributions. We intend to address this limitation in future research.

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

# A APPENDIX

## A.1 ONE-SIDED INTERVALS AND ASYMMETRIC INTERVALS

One-sided conformalised JPRs can be formed with a slight change to the conformity score. To do so we consider signed residuals instead of absolute ones at each time step, redefining the conformity score and present them in the multiple time series setting

$$\alpha_{i,+}^k = K\text{-}\max \left\{ \frac{y_{i,1} - \hat{y}_{i,1}}{\hat{\sigma}_1(X_i)}, \dots, \frac{y_{i,H} - \hat{y}_{i,H}}{\hat{\sigma}_H(X_i)} \right\}$$

for $i \in \{1, \dots, n_{cal}\}$ and where $K\text{-}\max(\vec{x})$ as the $K^{th}$ largest element of a sequence $\vec{x}$. One-sided lower JPRs for test sample $Z^* = (X^*, Y^*)$ can be given by

$$\mathcal{C}_{1-\epsilon}^{\alpha+} = \left( \hat{y}_1 - q_{1-\epsilon}^{K\text{-}\max} \cdot \hat{\sigma}_1(X^*), \infty \right) \times \dots \times \left( \hat{y}_H - q_{1-\epsilon}^{K\text{-}\max} \cdot \hat{\sigma}_H(X^*), \infty \right)$$

and analogously a one-side upper JPR is given by

$$\mathcal{C}_{1-\epsilon}^{\alpha-} = \left( -\infty, \hat{y}_1 + q_{1-\epsilon}^{K\text{-}\max} \cdot \hat{\sigma}_1(X^*) \right) \times \dots \times \left( -\infty, \hat{y}_H + q_{1-\epsilon}^{K\text{-}\max} \cdot \hat{\sigma}_H(X^*) \right).$$

where, $X^*$ are the lagged values(i.e the history) for the unknown $Y^*$ that is to be predicted.

We define $q_\eta^{K\text{-}\max}$ as the $\eta^{th}$ empirical quantile of the distribution of $\alpha_{i,+}^k$.

Asymmetric intervals: If the intervals are unlikely to be symmetric, one can adapt the inversion of quantiles from the conformity scores such as $\epsilon_+, \epsilon_- > 0$ with $\epsilon_+ + \epsilon_- = \epsilon$ thus the JPR is

$$\mathcal{C}_{1-\epsilon}^{\alpha\pm} = \left( \left( \hat{y}_1 - q_{1-\epsilon_-}^{K\text{-}\max} \cdot \hat{\sigma}_1(X^*) \right), \left( \hat{y}_1 + q_{1-\epsilon_+}^{K\text{-}\max} \cdot \hat{\sigma}_1(X^*) \right) \right) \times \dots$$

$$\dots \times \left( \left( \hat{y}_H - q_{1-\epsilon_-}^{K\text{-}\max} \cdot \hat{\sigma}_H(X^*) \right), \left( \hat{y}_H + q_{1-\epsilon_+}^{K\text{-}\max} \cdot \hat{\sigma}_H(X^*) \right) \right).$$

## A.2 DETAILS ON THEOREM 1

### A.2.1 ASSUMPTIONS FOR THEOREM 1

Let $\alpha^o$ be an oracle non-conformity measure, and let $\alpha$ be the corresponding non-conformity score for approximate results. Assume the number of randomisations, $d$, and the size of the training sequence, $m$, grow arbitrarily large, i.e. $d, m \to \infty$. Further, let $\{\delta_{1d}, \delta_{2m}, \gamma_{1d}, \gamma_{2m}\}$ be sequences of non-negative numbers converging to zero. We impose the following conditions:

(1) **Approximate Ergodicity:** With probability $1 - \gamma_{1d}$, the randomisation distribution

$$\hat{F}(x) := \frac{1}{d} \sum_{\pi \in \Pi} \mathbf{1}\{\alpha^o(Z_{cal}^\pi) < x\} \tag{9}$$

is approximately ergodic for

$$F(x) = \Pr(\alpha^o(Z_{cal}) < x) \tag{10}$$

that is $\sup_{x \in \mathbb{R}} |\hat{F}(x) - F(x)| \leq \delta_{1d}$;

(2) **Small estimation errors:** With probability $1 - \gamma_{2m}$ the following hold:
  (a) the mean squared error is small, $d^{-1} \sum_{\pi \in \Pi} [(\alpha(Z_{cal}^\pi) - \alpha^o(Z_{cal}^\pi))]^2 \leq \delta_{2m}^2$;
  (b) the pointwise error when $\pi$ is the identity permutation is small, $|\alpha(Z_{cal}) - \alpha^o(Z_{cal})| \leq \delta_{2m}$;
  (c) the pdf of $\alpha^o(Z_{cal})$ is bounded above by a constant $D$.

Condition (1) states an ergodicity condition, that permuting the oracle conformity scores provides a meaningful approximation to the unconditional distribution of the oracle conformity score. This holds when a time series is strongly mixing ($\alpha$-mixing) (7; 31; 7; 31) using the permutation scheme

discussed 4.2. Notably ARMA (Autoregressive and Moving Average) series with IID innovations are known to be $\alpha$-mixing (7; 31). Condition (2) bounds the discrepancy between the non-conformity scores and their oracle counterparts.

**Theorem 1 (Approximate General Validity of Inductive Conformal Inference)**

Under mild assumptions on ergodicity and small errors (see Appendix A.2.1), for any $\epsilon \in (0,1)$, the approximation of conformal p-value is approximately distributed as follows:

$$|\Pr(\hat{p} \leq \epsilon) - \epsilon| \leq 6\delta_{1d} + 2\delta_{2m} + 2D(\delta_{2m} + 2\sqrt{\delta_{2m}}) + \gamma_{1d} + \gamma_{2m} \tag{11}$$

for any $\epsilon \in (0,1)$ and the corresponding conformal set has an approximate coverage $1 - \epsilon$, i.e,

$$|\Pr(y^* \in \mathcal{C}_{1-\epsilon}) - (1 - \epsilon))| \leq 6\delta_{1d} + 2\delta_{2m} + 2D(\delta_{2m} + 2\sqrt{\delta_{2m}}) + \gamma_{1d} + \gamma_{2m}. \tag{12}$$

*Proof.* The proof largely follows the proof of Generalised Conformal Prediction in (9), adapted for Inductive Conformal Prediction. Since the second condition (bounds on the coverage probability) is implied by the first condition, it suffices to prove the first claim. Define the empirical distribution function of the non-conformity scores under randomization as:

$$\hat{F}(x) := \frac{1}{d} \sum_{\pi \in \Pi} \mathbf{1}\{\alpha^o(Z_{cal}^\pi) < x\} \tag{13}$$

The rest of the proof proceeds in two steps. We first bound $\hat{F}(x) - F(x)$ and then derive the desired result.

**Step 1:** We bound the difference between the $p$-value and the oracle $p$-value, $\hat{F}(\alpha(Z_{cal})) - F(\alpha^o(Z_{cal}))$. Let $\mathcal{M}$ be the event that the conditions (1) and (2) hold. By assumption,

$$\Pr(\mathcal{M}) \geq 1 - \gamma_{1d} - \gamma_{2m}. \tag{14}$$

Notice that on the event $\mathcal{M}$,

$$
\begin{aligned}
\left|\hat{F}(\alpha(Z_{cal})) - F(\alpha^o(Z_{cal}))\right| &\leq \left|\hat{F}(\alpha(Z_{cal})) - F(\alpha(Z_{cal}))\right| + |F(\alpha(Z_{cal}) - F(\alpha^o(Z_{cal}))| \\
&\overset{(i)}{\leq} \sup_{x \in \mathbb{R}} \left|\hat{F}(x) - F(x)\right| + D\left|\alpha(Z_{cal}) - \alpha^o(Z_{cal})\right| \\
&\leq \sup_{x \in \mathbb{R}} \left|\hat{F}(x) - \tilde{F}(x)\right| + \sup_{x \in \mathbb{R}} \left|\tilde{F}(x) - F(x)\right| + D|\alpha(Z_{cal}) - \alpha^o(Z_{cal})| \\
&\leq \sup_{x \in \mathbb{R}} \left|\hat{F}(x) - \tilde{F}(x)\right| + \delta_{1d} + D|\alpha(Z_{cal}) - \alpha^o(Z_{cal})| \\
&\leq \sup_{x \in \mathbb{R}} \left|\hat{F}(x) - \tilde{F}(x)\right| + \delta_{1d} + D\delta_{2m}, \tag{15}
\end{aligned}
$$

where (i) holds by the fact that the bounded pdf of $\alpha^o(Z_{cal})$ implies the Lipschitz property for $F$.

Let $A = \{\pi \in \Pi \colon |\alpha(Z_{cal}^\pi) - \alpha^o(Z_{cal}^\pi)| \geq \sqrt{\delta_{2m}}$. Notice that on the event $\mathcal{M}$, by Chebyshev inequality

$$|A|\delta_{2d} \leq \sum_{\pi \in \Pi} \left(\alpha(Z_{cal}^\pi) - \alpha^o(Z_{cal}^\pi)\right)^2 \leq d\delta_{2m}^2$$

and thus $|A|/m \leq \delta_{2m}$. Also notice that on the event $\mathcal{M}$, for any $x \in \mathbb{R}$,

$$\left| \hat{F}(x) - \tilde{F}(x) \right|$$

$$\leq \frac{1}{d} \sum_{\pi \in A} |\mathbf{1}\{\alpha(Z_{cal}^{\pi}) < x\} - \mathbf{1}\{\alpha^o(Z_{cal}^{\pi}) < x\}| + \frac{1}{d} \sum_{\pi \in (\Pi \backslash A)} |\mathbf{1}\{\alpha(Z_{cal}^{\pi}) < x\} - \mathbf{1}\{\alpha^o(Z_{cal}^{\pi}) < x\}|$$

$$\overset{(i)}{\leq} \frac{|A|}{d} + \frac{1}{d} \sum_{\pi \in (\Pi \backslash A)} \mathbf{1}\left\{|\alpha^o(Z_{cal}^{\pi}) - x| \leq \sqrt{\delta_2}\right\}$$

$$\leq \frac{|A|}{d} + \frac{1}{d} \sum_{\pi \in \Pi} \mathbf{1}\left\{|\alpha^o(Z_{cal}^{\pi}) - x| \leq \sqrt{\delta_{2m}}\right\}$$

$$\leq \frac{|A|}{d} + \Pr\left(|\alpha^o(Z_{cal}) - x| \leq \sqrt{\delta_{2m}}\right)$$

$$\qquad + \sup_{z \in \mathbb{R}} \left| \frac{1}{d} \sum_{\pi \in \Pi} \mathbf{1}\left\{|\alpha^o(Z_{cal}^{\pi}) - z| \leq \sqrt{\delta_{2m}}\right\} - \Pr\left(|\alpha^o(Z_{cal}) - z| \leq \sqrt{\delta_{2m}}\right)\right|$$

$$= \frac{|A|}{d} + \Pr\left(|\alpha^o(Z_{cal}) - x| \leq \sqrt{\delta_{2m}}\right)$$

$$\qquad + \sup_{x \in \mathbb{R}} \left|\left[\tilde{F}\left(z + \sqrt{\delta_{2m}}\right) - \tilde{F}\left(z - \sqrt{\delta_{2m}}\right)\right] - \left[F\left(z + \sqrt{\delta_{2m}}\right) - F\left(z - \sqrt{\delta_{2m}}\right)\right]\right|$$

$$\leq \frac{|A|}{d} + \Pr\left(|\alpha^o(Z_{cal}) - x| \leq \sqrt{\delta_{2m}}\right) + 2 \sup_{x \in \mathbb{R}} \left|\tilde{F}(z) - F(z)\right|$$

$$\overset{(ii)}{\leq} \frac{|A|}{d} + 2D\sqrt{\delta_{2m}} + 2\delta_{1d}$$

$$\overset{(iii)}{\leq} 2\delta_{1d} + \delta_{2m} + 2D\sqrt{\delta_{2m}},$$

where

    i. follows by the boundedness of indicator functions and the elementary inequality of $|\mathbf{1}\{\alpha(Z_{cal}^{\pi}) < x\} - \mathbf{1}\{\alpha^o(Z_{cal}^{\pi}) < x\}| \leq \mathbf{1}\{|\alpha^o(Z_{cal}^{\pi}) - x| \leq |\alpha(Z_{cal}^{\pi}) - \alpha^o(Z_{cal}^{\pi})|\}$;

    ii. follows by the bounded pdf of $\alpha^o(Z_{cal})$;

    iii. follows by $|A|/d \leq \delta_{2m}$.

Since the above result holds for each $x \in \mathbb{R}$, it follows that on the event $\mathcal{M}$,

$$\sup_{x \in \mathbb{R}} \left|\hat{F}(x) - \tilde{F}(x)\right| \leq 2\delta_{1d} + \delta_{2m} + 2D\sqrt{\delta_{2m}}. \tag{16}$$

We combine (15) and (16) and obtain that on the event $\mathcal{M}$,

$$\left|\hat{F}(\alpha(Z_{cal})) - F(\alpha^o(Z_{cal}))\right| \leq 3\delta_{1d} + \delta_{2m} + D(\delta_{2m} + 2\sqrt{\delta_{2m}}). \tag{17}$$

**Step 2:** The derivation of the main result follows: Notice that

$$\left| \Pr\left(1 - \hat{F}(\alpha(Z_{cal})) \leq \epsilon\right) - \epsilon \right|$$

$$= \left| E\left(\mathbf{1}\left\{1 - \hat{F}(\alpha(Z_{cal})) \leq \epsilon\right\} - \mathbf{1}\{1 - F(\alpha^o(Z_{cal})) \leq \epsilon\}\right)\right|$$

$$\leq E\left|\mathbf{1}\left\{1 - \hat{F}(\alpha(Z_{cal})) \leq \epsilon\right\} - \mathbf{1}\left\{1 - F(\alpha^o(Z_{cal})) \leq \epsilon\right\}\right|$$

$$\overset{(i)}{\leq} \Pr\left(|F(\alpha^o(Z_{cal})) - 1 + \epsilon| \leq \left|\hat{F}(\alpha(Z_{cal})) - F(\alpha^o(Z_{cal}))\right|\right)$$

$$\leq \Pr\left(|F(\alpha^o(Z_{cal})) - 1 + \epsilon| \leq \left|\hat{F}(\alpha(Z_{cal})) - F(\alpha^o(Z_{cal}))\right| \text{ and } \mathcal{M}\right) + \Pr\left(\mathcal{M}^{\mathcal{C}}\right)$$

$$\overset{(ii)}{\leq} \left(\Pr(|F(\alpha^o(Z_{cal})) - 1 + \epsilon| \leq 3\delta_{1d} + \delta_{2m} + D(\delta_{2m} - 2\sqrt{\delta_{2m}}))\right) + \Pr(\mathcal{M}^{\mathcal{C}})$$

$$\overset{(iii)}{\leq} 6\delta_{1d} + 2\delta_{2m} + 2D(\delta_{2m} - 2\sqrt{\delta_{2m}}) + \gamma_{1d} + \gamma_{2m},$$

where

    i. follows by the elementary inequality $|\mathbf{1}\{1 - \hat{F}(\alpha(Z_{cal})) \leq \epsilon\} - \mathbf{1}\{1 - F(\alpha^o(Z_{cal})) \leq \epsilon\}| \leq |\mathbf{1}\{F(\alpha^o(Z_{cal})) - 1 + \epsilon| \leq |\hat{F}(\alpha(Z_{cal})) - F(\alpha^o(Z_{cal}))|\}$,

    ii. follows by (17),

    iii. follows by the fact that $F(\alpha^o(Z_{cal}))$ has the uniform distribution on $(0, 1)$ and therefore, has pdf equal to 1, and by (14).

$\square$

**Theorem 2** (General Exact Validity). *Consider a sequence of observations $\mathbf{Z}^\pi_{cal}$ that has an exchangeable distribution under the permutation group $\Pi$. For any fixed permutation group the randomisation quantiles, $\epsilon$, are invariant, namely*

$$\mathcal{A}^{(r(\epsilon))}\left(Z_{tr}, Z^\pi_{cal}\right) = \mathcal{A}^{(r(\epsilon))}\left(Z_{tr}, Z_{cal}\right) \ \forall \pi \in \Pi$$

*where $r(\epsilon) = \lceil(d+1)\epsilon)\rceil$-th non-conformity score (when ranked in the descending order). Then, the following probabilistic guarantees hold:*

- $\Pr(\hat{p} \leq \epsilon) = \Pr\left(\mathcal{A}\left(Z_{tr}, Z_{cal}\right) > \mathcal{A}\left(Z_{tr}, Z^{(r(\epsilon))}_{cal}\right)\right) \leq \epsilon$

- $\Pr(Y^* \in \mathcal{C}_{1-\epsilon}) \geq 1 - \epsilon$

*where $Z^{(r(\epsilon))}_{cal}$ is the permuted sequence that corresponds to the $\lceil(d+1)\epsilon\rceil^{th}$ non-conformity score when ranked in descending order, $\mathcal{C}_{1-\epsilon}$ is the prediction interval for $(1 - \epsilon)$ coverage and test sample $Z^* = (X^*, Y^*)$.*

The proof follows from arguments for randomisation inference that can be found in (9; 32).

## B TRAINING DETAILS

We perform all our experiments on Intel(R) Xeon(R) W-2265 CPU @ 3.50GHz with 20 CPUs, 12 cores per socket, and 2 threads per core. In totality, we used on the order of 100 compute hours.

### B.1 SINGLE TIME SERIES EXPERIMENTS

For Monte Carlo simulations and the US GDP dataset, we train AR(2) models as the main predictor. For learning scaling factors, we use a linear regression model and use the last 6 steps as the features to output the scaling factors.

### B.2 PARTICLE EXPERIMENTS

We take the same experimental setup as (36) and use a 1-layer sequence-to-sequence LSTM network (EncDec) where the encoder has a single LSTM layer with embedding size 24 and the decoder has a single LSTM layer with embedding size 24 and a linear layer. We also fit a 1-layer RNN with a single LSTM layer (RNN) with embedding size 24, followed by a linear layer. We train the model for 150 epochs and set batch size to 150.

For each dataset, 5000 samples were generated and split into 45/45/10 proportions for training, calibration, and testing, respectively. Baselines not requiring calibration used the calibration split for training.

### B.3 UK COVID EXPERIMENTS

We take the same architectures from the particle experiments and apply them to the same COVID data as (33). We train these models for 200 epochs with embedding sizes of 128 and set batch size of 64.

Of the 380 time series, we utilise 200 sequences for training, 100 for calibration, and 80 for testing.

### B.4 COMPLETE MONTE CARLO RESULTS

We visualise the results of these experiments in Figure 2. For complete numerical results see Tables 3-8. We find that our **JANET** methods are comparable or better with respect to coverage and interval widths against the baselines.

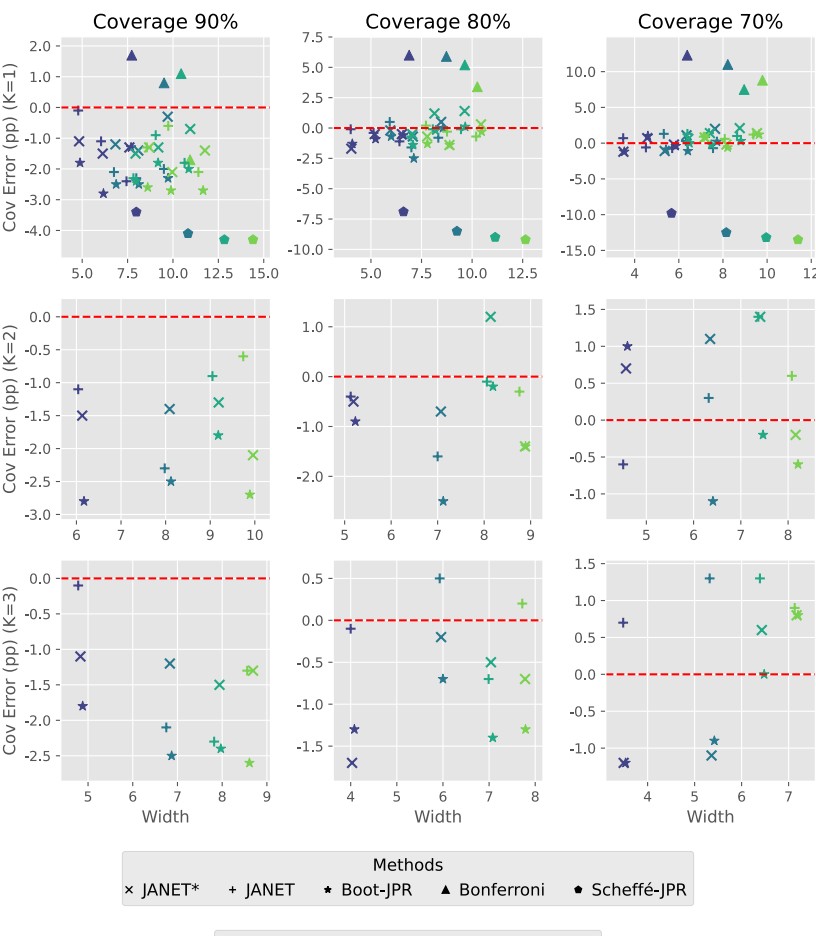

Figure 2: Monte Carlo Experiment Coverage Error (pp) vs Interval Width. The $y$-axis shows the difference in coverage from the target $1 - \epsilon$ for $\epsilon = 0.2$ in percentage points and the $x$-axis is the geometric mean of the interval width over the time horizon. The red line represents perfect calibration. Better methods can be found near the red line and to the left (well-calibrated, narrower interval width). The left plot is for the case of $K = 1$ while the right plot shows $K = 3$. The Bonferroni and Scheffé-JPR methods are only applicable for $K = 1$. Shapes represent calibration methods and colors signify forecast horizon, $H$. For $K = 1$ (top row) note that the Bonferroni regions are consistently overconservative (overcoverage) while the Scheffé regions are consistently anticonservative (undercoverage). Meanwhile the bootstrap and **JANET** JPRs are comparable and generally provide close to the desired coverage and are similar in width.

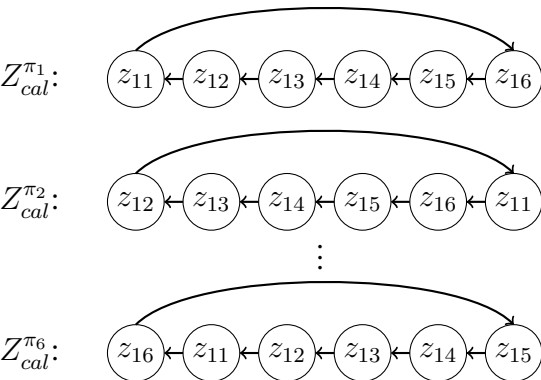

Figure 3: Visualisation of NOB permutations applied to a single time series $Z$ of length $L = L_{tr} + L_{cal} = 10 + 6 = 16$ ($z_1, \ldots, z_{10}$ are used for training and $z_{11}, \ldots, z_{16}$ are reserved for calibration). The first row shows the calibration portion of a time series, $Z$, under the identity permutation, $\pi_1$. Subsequent rows show how different permutations rearrange $Z$ for block size, $b = 1$. Arrows denote how the objects are permuted. Each permutation in this group rotates the front block to the end.

Table 3: Empirical Coverages of Monte Carlo Simulations with Significance Level $\epsilon = 0.1$. Coverages are presented as percentages.

|  | $\epsilon = 0.1$ | | | |
| --- | --- | --- | --- | --- |
| Method | $H = 6$ | $H = 12$ | $H = 18$ | $H = 24$ |
| Bonferroni | 91.7 | 90.8 | 91.1 | 88.3 |
| Scheffé-JPR | 86.6 | 85.9 | 85.7 | 85.7 |
| Boot-JPR($K = 1$) | 88.7 | 87.7 | 88.0 | 87.3 |
| **JANET**$^*$($K = 1$) | 87.6 | 88.0 | 88.2 | 87.9 |
| **JANET**($K = 1$) | 88.7 | 89.7 | 89.3 | 88.6 |
| Boot-JPR($K = 2$) | 87.2 | 87.5 | 88.2 | 87.3 |
| **JANET**$^*$($K = 2$) | 88.9 | 87.7 | 89.1 | 89.4 |
| **JANET**($K = 2$) | 88.5 | 88.6 | 88.7 | 87.9 |
| Boot-JPR($K = 3$) | 88.2 | 87.5 | 87.6 | 87.4 |
| **JANET**$^*$($K = 3$) | 89.9 | 87.9 | 87.7 | 88.7 |
| **JANET**($K = 3$) | 88.9 | 88.8 | 88.5 | 88.7 |

Table 4: Empirical Coverages of Monte Carlo Simulations with $\epsilon = 0.2$. Coverages are presented as percentages. The desired coverage is 80%.

|  | $\epsilon = 0.2$ | | | |
| --- | --- | --- | --- | --- |
| Method | $H = 6$ | $H = 12$ | $H = 18$ | $H = 24$ |
| Bonferroni | 86.0 | 85.9 | 85.2 | 83.4 |
| Scheffé-JPR | 73.1 | 71.5 | 71.0 | 70.8 |
| Boot-JPR($K = 1$) | 79.4 | 80.0 | 80.1 | 79.7 |
| **JANET**$^*$($K = 1$) | 78.9 | 79.2 | 79.9 | 79.3 |
| **JANET**($K = 1$) | 79.5 | 80.5 | 81.4 | 80.3 |
| Boot-JPR($K = 2$) | 79.1 | 77.5 | 79.8 | 78.6 |
| **JANET**$^*$($K = 2$) | 79.6 | 78.4 | 79.9 | 79.7 |
| **JANET**($K = 2$) | 79.5 | 79.3 | 81.2 | 78.6 |
| Boot-JPR($K = 3$) | 78.7 | 79.3 | 78.6 | 78.7 |
| **JANET**$^*$($K = 3$) | 79.9 | 80.5 | 79.3 | 80.2 |
| **JANET**($K = 3$) | 78.3 | 79.8 | 79.5 | 79.3 |

Table 5: Empirical Coverages of Monte Carlo Simulations with $\epsilon = 0.3$. Coverages are presented as percentages. Desired coverage is 70%.

| Method | $\epsilon = 0.3$ | | | |
|---|---|---|---|---|
| | $H = 6$ | $H = 12$ | $H = 18$ | $H = 24$ |
| Bonferroni | 82.3 | 81.0 | 77.5 | 78.8 |
| Scheffé-JPR | 60.2 | 57.5 | 56.8 | 56.5 |
| Boot-JPR($K = 1$) | 69.6 | 70.2 | 70.3 | 71.2 |
| **JANET**$^*$($K = 1$) | 69.3 | 69.3 | 71.0 | 71.2 |
| **JANET**($K = 1$) | 69.8 | 72.0 | 72.1 | 71.4 |
| Boot-JPR($K = 2$) | 71.0 | 68.9 | 69.8 | 69.4 |
| **JANET**$^*$($K = 2$) | 69.4 | 70.3 | 71.4 | 70.6 |
| **JANET**($K = 2$) | 70.7 | 71.1 | 71.4 | 69.8 |
| Boot-JPR($K = 3$) | 68.8 | 69.1 | 70.0 | 70.8 |
| **JANET**$^*$($K = 3$) | 70.7 | 71.3 | 71.3 | 70.9 |
| **JANET**($K = 3$) | 68.8 | 68.9 | 70.6 | 70.8 |

Table 6: Empirical Widths of Monte Carlo Simulations with $\epsilon = 0.1$. The desired coverage is 90% and narrower intervals are preferred. These reported widths are geometric means over the time steps averaged over the 1000 simulations.

| Method | $\epsilon = 0.1$ | | | |
|---|---|---|---|---|
| | $H = 6$ | $H = 12$ | $H = 18$ | $H = 24$ |
| Bonferroni | 7.73 | 9.51 | 10.45 | 10.94 |
| Scheffé-JPR | 7.98 | 10.83 | 12.83 | 14.41 |
| Boot-JPR(K=1) | 7.69 | 9.73 | 10.87 | 11.66 |
| **JANET**$^*$($K = 1$) | 7.44 | 9.50 | 10.64 | 11.40 |
| **JANET**($K = 1$) | 7.60 | 9.71 | 10.95 | 11.76 |
| Boot-JPR($K = 2$) | 6.17 | 8.12 | 9.18 | 9.89 |
| **JANET**$^*$($K = 2$) | 6.04 | 7.98 | 9.05 | 9.74 |
| **JANET**($K = 2$) | 6.13 | 8.09 | 9.19 | 9.96 |
| Boot-JPR($K = 3$) | 4.88 | 6.87 | 7.97 | 8.61 |
| **JANET**$^*$($K = 3$) | 4.78 | 6.75 | 7.82 | 8.56 |
| **JANET**($K = 3$) | 4.83 | 6.83 | 7.94 | 8.69 |

Table 7: Empirical Widths of Monte Carlo Simulations with $\epsilon = 0.2$. The desired coverage is 80%. Narrower intervals are preferred. These reported widths are geometric means over the time steps averaged over the 1000 simulations.

| Method | $\epsilon = 0.2$ | | | |
|---|---|---|---|---|
| | $H = 6$ | $H = 12$ | $H = 18$ | $H = 24$ |
| Bonferroni | 6.89 | 8.73 | 9.64 | 10.26 |
| Scheffé-JPR | 6.61 | 9.25 | 11.14 | 12.64 |
| Boot-JPR($K = 1$) | 6.59 | 8.56 | 9.66 | 10.47 |
| **JANET**$^*$($K = 1$) | 6.41 | 8.33 | 9.45 | 10.20 |
| **JANET**($K = 1$) | 6.53 | 8.47 | 9.63 | 10.44 |
| Boot-JPR($K = 2$) | 5.23 | 7.12 | 8.19 | 8.89 |
| **JANET**$^*$($K = 2$) | 5.13 | 7.00 | 8.06 | 8.76 |
| **JANET**($K = 2$) | 5.19 | 7.07 | 8.14 | 8.88 |
| Boot-JPR($K = 3$) | 4.08 | 6.00 | 7.08 | 7.79 |
| **JANET**$^*$($K = 3$) | 4.00 | 5.93 | 6.99 | 7.72 |
| **JANET**($K = 3$) | 4.03 | 5.96 | 7.04 | 7.78 |

Table 8: Empirical Widths of Monte Carlo Simulations with $\epsilon = 0.3$. The desired coverage is 70%. Narrower intervals are preferred. These reported widths are geometric means over the time steps averaged over the 1000 simulations.

| | $\epsilon = 0.3$ | | | |
| Method | $H = 6$ | $H = 12$ | $H = 18$ | $H = 24$ |
|---|---|---|---|---|
| Bonferroni | 6.38 | 8.22 | 8.96 | 9.79 |
| Scheffé-JPR | 5.67 | 8.15 | 9.96 | 11.40 |
| Boot-JPR($K = 1$) | 5.86 | 7.73 | 8.81 | 9.63 |
| **JANET***($K = 1$) | 5.70 | 7.55 | 8.64 | 9.38 |
| **JANET**($K = 1$) | 5.78 | 7.64 | 8.76 | 9.54 |
| Boot-JPR($K = 2$) | 4.60 | 6.41 | 7.47 | 8.21 |
| **JANET***($K = 2$) | 4.51 | 6.32 | 7.37 | 8.08 |
| **JANET**($K = 2$) | 4.57 | 6.35 | 7.41 | 8.16 |
| Boot-JPR($K = 3$) | 3.53 | 5.42 | 6.48 | 7.20 |
| **JANET***($K = 3$) | 3.48 | 5.32 | 6.39 | 7.13 |
| **JANET**($K = 3$) | 3.49 | 5.36 | 6.43 | 7.17 |

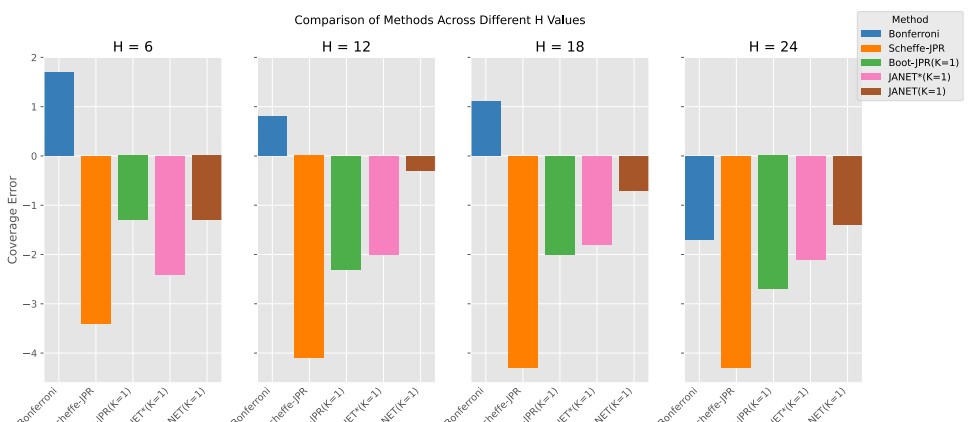

Figure 4: Bar plot for the coverages of different methods, $\epsilon = 0.1$. The desired coverage is 90%. Negative values indicate undercoverage and positive values indicate overcoverage.

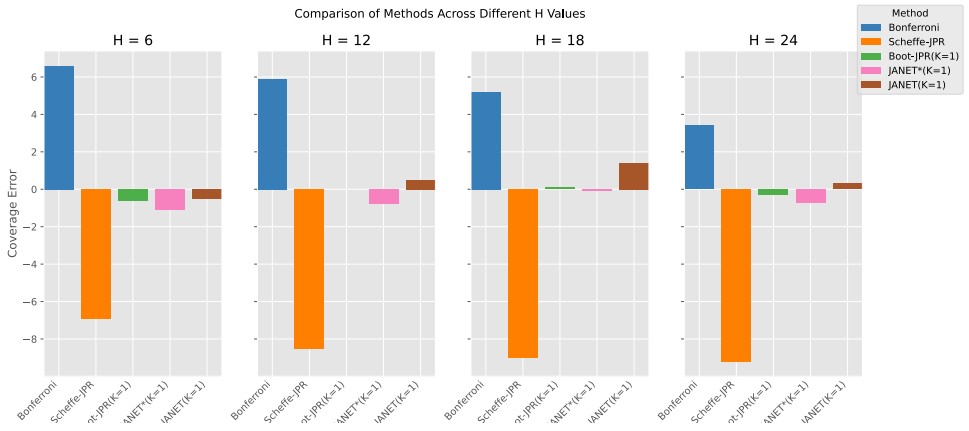

Figure 5: Bar plot for the coverages of different methods, $\epsilon = 0.2$ (80% coverage).

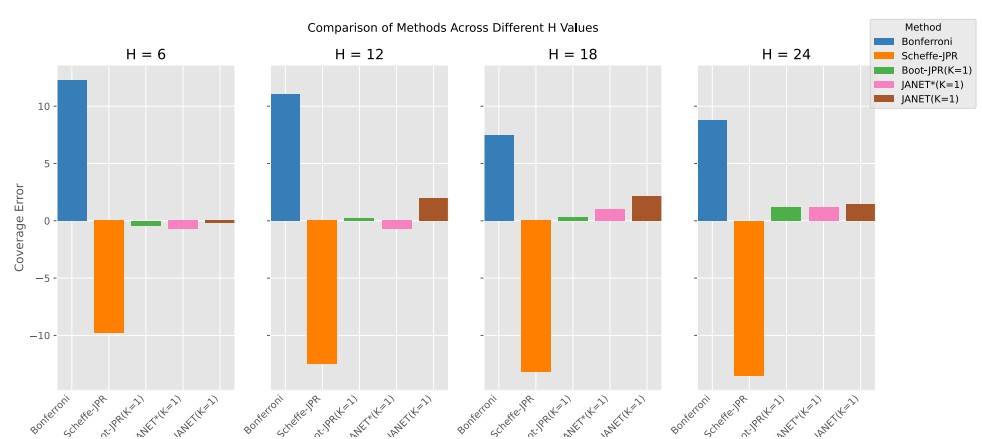

Figure 6: Bar plot for the coverages of different methods, $\epsilon = 0.3$ (70% coverage)

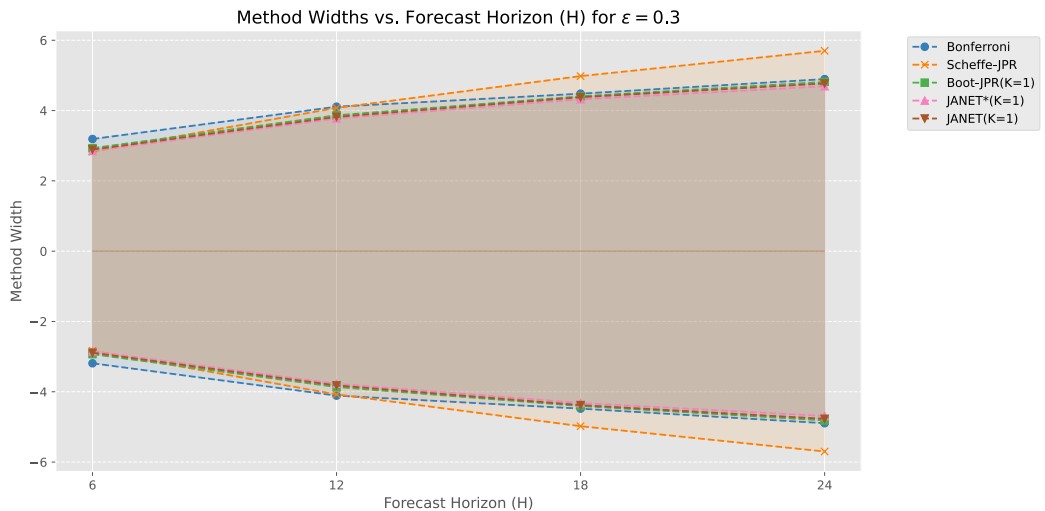

Figure 7: Geometric mean of widths for different forecast horizons of different methods, $\epsilon = 0.3$ (70% coverage).

s

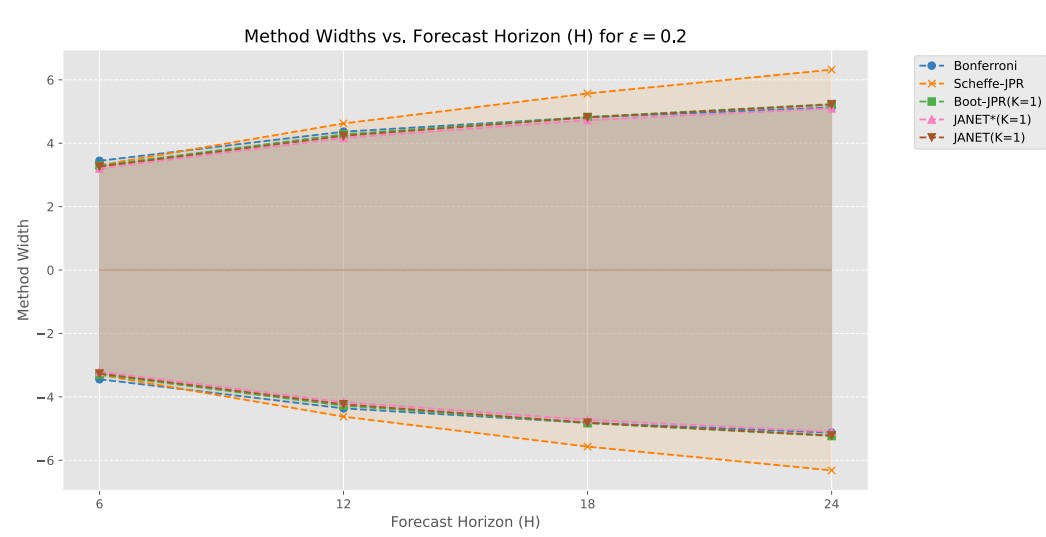

Figure 8: Geometric mean of widths for different forecast horizons of different methods, $\epsilon = 0.2$ (80% coverage).

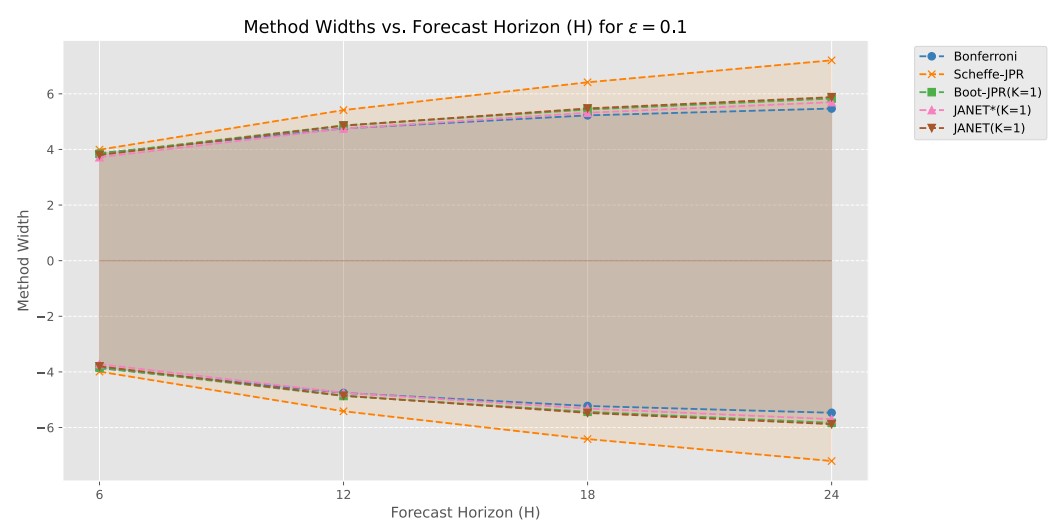

Figure 9: Geometric mean of widths for different forecast horizons of different methods, $\epsilon = 0.1$ (90% coverage).

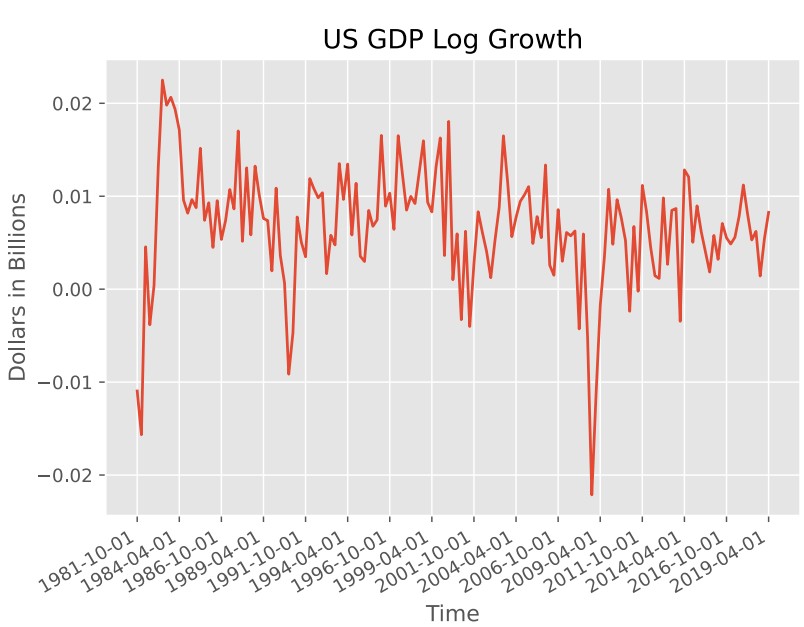

Figure 10: The resulting GDP data after preprocessing (log transform and differencing).

