# OpenReview forum: "JANET: Joint Adaptive predictioN-region Estimation for Time-series"
_ICLR.cc/2025/Conference — ICLR 2025 Conference Withdrawn Submission_

### Official Review · Reviewer_UZnm · 2024-10-17

**Soundness:** 3
**Presentation:** 2
**Contribution:** 3
**Rating:** 5
**Confidence:** 4

**Summary:**

This work considers the important problem of constructing multi-step prediction sets in time series with uncertainty control. They treat each time series with same horizen as a "data point" and design a novel non-conformity score to provide joint prediction region. Extensive experiments are conducted to demonstrate its effectiveness.

**Strengths:**

S1. The problem of constructing multi-step prediction regions is important in practice, and uncertainty quantification across multiple sets becomes more challenging due to multiplicity. The concept of K-FWER introduced in this paper is a reasonable approach to effectively quantify error.

S2. The paper addresses the conservativeness of existing methods for controlling K-FWER and proposes a direct approach to construct joint prediction regions using a well-designed non-conformity score. Empirical results also confirm the effectiveness of this method.

**Weaknesses:**

**Weakness 1.** The paper lacks a concrete theoretical discussion on K-FWER control for the proposed method. The relationship between the designed score and K-FWER control should be made explicit. While I understand the authors treat each time series with horizon H as a single data point and apply conformal inference accordingly, this approach may be unclear to new readers.

**Weakness 2.** For a single time series, the K-FWER control remains questionable. The authors should provide thorough theoretical justification, particularly in relation to Theorem 1.

**Weakness 3.** The background introduction is overly lengthy. Several sentences, such as those in lines 153-155 explaining $a_{(n_{cal})}$ appear redundant. The authors only present their approach in Section 5 which is quite late.  I suggest that after Section 3, the authors introduce their method by illustrating the case of multiple time series first, and then discuss the single time series as the extension.

**Questions:**

**Question 1**: In my opinion, under the ideal case of independent multiple time series, the proposed method only guarantees Pr (K of the H components are not in the computed region and H-K components are in the region)$\leq\alpha,$ which seems like a conservative version of K-FWER control. Is this correct? If not, could the authors provide detailed proofs to clarify this?

**Question 2**: How do the horizon size H and the  error tolerance K effects the K-FWER control for single time series. Can the authors present a bound for the coverage gap by applying Theorem 1 to this case?

**Question 3**: I noticed that JANET shows significant inflation in coverage in certain experiments, such as Table 2 (JANET-RNN, Particle1), where it reports 87% coverage without finite sample control. I think this is because the current experiments only report the "frequency of coverage" on the test set, which is not an unbiased estimator of K-FWER. I would like to see experiments verifying the finite sample K-FWER control of JANET for multiple independent time series by sufficient replications.

**Question 4**: The references are not properly cited. Many references lack information about the journals or conferences in which they were published. Please check and correct this.

---

### Official Review · Reviewer_zPnd · 2024-10-18

**Soundness:** 3
**Presentation:** 2
**Contribution:** 2
**Rating:** 3
**Confidence:** 3

**Summary:**

This paper presents a new algorithm for constructing joint prediction regions for multi-step time series prediction, JANET. The authors present results on both single time-series and multiple time-series benchmarks.

**Strengths:**

*Motivation*. The paper fills in the gap of generating multi-time-step joint prediction regions in the  case of only one time series exists. The authors demonstrated that method can also work well for the case when other independent time series exists for calibration.

*Soundness*. The theoretical results in the paper are well argued and sound.

**Weaknesses:**

## Clarity questions

- Definition 1 / Theorem 1. What is $\mathcal{A}$ and the various constants $\delta$s and $\gamma$s? They are undefined in the paper.  What does $\mathcal{A}$ it output, if it only takes two datasets as input and does not take in the test sample? Overall 4.3 was not clear to me. Isn't eq 3 and 4 the same equation with an reused $\epsilon$? Theorem 1 can benefit from some explanation as well (i.e. what does the error depend on, how it behaves with different data properties etc.)

- the notations are a bit messy throughout the paper. For example, what is the arrow x in line 299? (this sequence notation is not used anywhere else, you use bold $\mathbb{X}$ instead). in Eq 5 and 6 the $\hat{\sigma}_i$ can be either a function or a scalar.  In line 306, is $X$ the input at this time step or relevant history?  Cleaning up these notations will make the paper easier to follow.

- Line 315: What does "inverting the conformity scores" mean?

##  Writing / Presentation

- deviation-scaled conformal intervals are only "adaptive" in a very narrow sense (of X*'s deviation from other training data) and doesn't always present better efficiency. In times series CP literature, adaptivity usually means adaptive to changes in nonconformity score distribution (along the lines of ACI or CAHFT).

- From my understanding, the paper's methodology contains two parts: permutation for calibration, and scaling. Both of them are existing CP algorithms adapted by the authors to the time series setting, but it is unclear what are the authors' modifications. Can you add a few sentences to each of the two sections on what are your original contributions?

- A similar concern as above, I fear that the setting of the method is not well explained. The author repeatedly emphasized that the algorithm's guarantees hold "under mild assumptions", but in reality the data has to be strongly mixing and have ergodicity. I would recommend the authors to further explain in the paper the assumptions being made, and have examples of the kind of data the algorithm is suitable for.


## to summarize

I believe the paper addresses an well-motivated problem and presents a sound method, but needs significant restructuring and clarification to be at the standard of a conference like ICLR. If these concerns are addressed, I'm happy to increase my score.

**Questions:**

see weaknesses.

---

### Official Review · Reviewer_YX3e · 2024-11-04

**Soundness:** 2
**Presentation:** 2
**Contribution:** 2
**Rating:** 3
**Confidence:** 4

**Summary:**

This paper proposes JANET, a conformal framework designed to generate joint prediction regions for multi-step forecasting with time series data. The primary contribution lies in the introduction of a non-conformity score defined as the $K$-max of the locally weighted residuals ([2]) over forecast steps, enabling control of the relaxed familywise error rate. The framework then leverages Generalized (split) Conformal Prediction ([1]) to produce the joint prediction regions.

[1] Victor Chernozhukov, Kaspar Wüthrich, and Zhu Yinchu. "Exact and robust conformal inference methods for predictive machine learning with dependent data." In *Proceedings of the 31st Conference On Learning Theory*. PMLR, 2018.

[2] Lei, Jing, et al. "Distribution-free predictive inference for regression." *Journal of the American Statistical Association* 113.523 (2018): 1094-1111.

**Strengths:**

The authors propose a non-conformity score that effectively controls the *relaxed* familywise error rate for multi-step prediction in an offline setting. This is a clever idea to prevent the excessive conservatism that can result from controlling the exact FWER, which often leads over-coverage.

**Weaknesses:**

- It is well-known that split conformal is a special case of full conformal([3], also highlighted directly in [1]). While split conformal offers computational advantages due to requiring only one model fitting, the motivation for framing the discussion solely from the split conformal perspective seems lacking when a more general approach could be described.
- The theoretical results are identical to those presented in [1]. A more thorough discussion differentiating this work from [1] would strengthen the paper’s contribution.
- Although the authors emphasize the applicability of their method to multivariate time series, all experiments appear to be conducted solely on univariate data.
- More detailed discussion is needed on whether the method provides significant performance gains compared to current state-of-the-art methods. Additionally, while the results suggest slight under-coverage in many cases, it is unclear if the width reduction compensates adequately for this under-coverage.
- The authors place significant emphasis on computational efficiency; however, this advantage seems to stem from the use of split conformal rather than the proposed algorithm itself. Moreover, despite this emphasis, the analysis of computational complexity is limited to a brief mention in Section 6.1.2.

Typos:
- Line 222: $l \to L_{cal}$

[3] Barber, Rina Foygel, et al. "Conformal prediction beyond exchangeability." *The Annals of Statistics* 51.2 (2023): 816-845.

**Questions:**

Please see Weaknesses.

+ Would the pre-processing of distinctly non-stationary data, such as the COVID-19 cases in the paper, critically impact performance?

---

### Official Review · Reviewer_A4YV · 2024-11-04

**Soundness:** 3
**Presentation:** 3
**Contribution:** 2
**Rating:** 3
**Confidence:** 4

**Summary:**

The paper presents a promising framework, JANET, for joint prediction regions in time-series analysis, but it falls short in critical areas that limit its contribution to the field. The theoretical results are somewhat incremental and do not substantively advance beyond current methodologies, reducing the novelty of the work. Additionally, the lack of comparisons with state-of-the-art approaches in multi-dimensional conformal time-series analysis weakens the empirical evaluation, particularly as one-step-ahead prediction—a crucial baseline—has been omitted. Finally, the literature review is incomplete, missing key recent works that provide valuable context and benchmarks for conformal prediction in time-series settings. These limitations suggest that the paper does not yet reach the level of rigor and comprehensiveness required for acceptance.

**Strengths:**

The paper introduces JANET (Joint Adaptive Prediction-region Estimation for Time-series), a framework designed to construct joint prediction regions (JPRs) for time-series prediction, accommodating both univariate and multivariate data, to multi-step predictions in time series where such assumptions do not hold. JANET offers guarantees by leveraging an inductive conformal prediction (ICP) approach, which only requires a single model fit. Key contributions include adapting non-conformity scores for multi-dimensional residuals, controlling K-familywise error rates (allowing specified error tolerance in predictions), and incorporating historical context to improve prediction accuracy.

The novelty lies in the generalization of CP to multi-dimensional time-series data with controlled K-familywise error rates. JANET extends prior work on CP by allowing joint prediction regions across multiple time steps and accounting for temporal dependencies through adaptive non-conformity scores. This advancement addresses the limitations of standard CP for multi-step predictions and offers approximate validity under certain ergodicity assumptions, making it applicable to single and independent time-series data. The experimental evaluation covers both single and multiple time-series settings, with JANET demonstrating competitive performance against several baseline methods and there are simulations on synthetic and real datasets (e.g., GDP data and COVID-19 case counts).

**Weaknesses:**

However, my concerns are as follows
- The theoretical results are a bit limited and incremental compared with existing results;

- There are not enough comparisons with state-of-the-art methods in multi-dimensional conformal time-series analysis, such as [42] published in ICML 2024. I believe the method should at least be compared with respect to one-step ahead prediction, since it is a special case of multi-step prediction. There is no fundamental difference in constructing multi-step ahead prediction from single step ahead, if you treat the prediction for y_{t+s}, s > 1, as the response variable instead of y_{t+1}.

-There are many conformal prediction for time series paper as of now in the literature, and I feel the literature survey is incomplete and missing several key references, such as

Angelopoulos, Anastasios, Emmanuel Candes, and Ryan J. Tibshirani. "Conformal pid control for time series prediction." Advances in neural information processing systems 36 (2024).

Yang, Zitong, Emmanuel Candès, and Lihua Lei. "Bellman Conformal Inference: Calibrating Prediction Intervals For Time Series." arXiv preprint arXiv:2402.05203 (2024).

**Questions:**

Could authors add additional experimental comparison, as well as better explaining the difference from existing literature?

---

### Note · Authors · 2024-11-24

I have read and agree with the venue's withdrawal policy on behalf of myself and my co-authors.